# Isolating and cryopreserving pig skin cells for single-cell RNA sequencing study

Li Han[1,2,3,4☯], Carlos P. Jara[5,6☯], Ou Wang[2], Yu Shi[1], Xinran Wu[3,4], Sandra Thibivilliers[7], Rafał K. Wóycicki[7], Mark A. Carlson[8,9], William H. Velander[2], Eliana P. Araújo[5,6], Marc Libault[7]*, Chi Zhang[1]*, Yuguo Lei[2,3,4,8,9,10]*

1 School of Biological Science, University of Nebraska, Lincoln, Nebraska, United States of America,
2 Department of Chemical and Biomolecular Engineering, University of Nebraska, Lincoln, Nebraska, United States of America, 3 Department of Biomedical Engineering, Pennsylvania State University, University Park, Pennsylvania, United States of America, 4 Huck Institutes of the Life Sciences, Pennsylvania State University, University Park, Pennsylvania, United States of America, 5 Nursing School, University of Campinas, Campinas SP, Brazil, 6 Laboratory of Cell Signaling, University of Campinas, Campinas SP, Brazil, 7 Department of Agronomy and Horticulture, Center for Plant Science Innovation, University of Nebraska, Lincoln, Nebraska, United States of America, 8 Mary and Dick Holland Regenerative Medicine Program, University of Nebraska Medical Center, Omaha, Nebraska, United States of America, 9 Department of Surgery, University of Nebraska Medical Center and the VA Nebraska-Western Iowa Health Care System, Omaha, Nebraska, United States of America, 10 Sartorius Mammalian Cell Culture Facility, Pennsylvania State University, University Park, Pennsylvania, United States of America

☯ These authors contributed equally to this work.
* marc.libault@unl.edu (ML); zhang.chi@unl.edu (CZ); yxl6034@psu.edu (YL)

**Data Availability Statement:** scRNA-seq data have been deposited in the GEO database under accession code GSE166561. The codes and walkthroughs are available at https://github.com/blinkeado/pigskin.

## Abstract

The pig skin architecture and physiology are similar to those of humans. Thus, the pig model is very valuable for studying skin biology and testing therapeutics. The single-cell RNA sequencing (scRNA-seq) technology allows quantitatively analyzing cell types, compositions, states, signaling, and receptor-ligand interactome at single-cell resolution and at high throughput. scRNA-seq has been used to study mouse and human skins. However, studying pig skin with scRNA-seq is still rare. A critical step for successful scRNA-seq is to obtain high-quality single cells from the pig skin tissue. Here we report a robust method for isolating and cryopreserving pig skin single cells for scRNA-seq. We showed that pig skin could be efficiently dissociated into single cells with high cell viability using the Miltenyi Human Whole Skin Dissociation kit and the Miltenyi gentleMACS Dissociator. Furthermore, the obtained single cells could be cryopreserved using 90% FBS + 10% DMSO without causing additional cell death, cell aggregation, or changes in gene expression profiles. Using the developed protocol, we were able to identify all the major skin cell types. The protocol and results from this study are valuable for the skin research scientific community.

## Introduction

Millions of people are affected by skin injuries and diseases [1–3]. Animal models are widely used to understand skin physiopathology and to test potential therapeutics [4,5]. Among various experimental animals, the skin architecture and physiology of pigs are closest to humans

**Funding:** This research was supported by the University of Nebraska Research Initiative 2018-2019 (YL and WV); the Nebraska DHHS Stem Cell Grant 2019 (YL and WV); National Cancer Institute grant 1R33CA235326 (YL); the U.S. Army GRANT10824516 (WV); the Pennsylvania State University start-up (YL); the Department of Defense, USA, W81XWH-BAA-11-1 (WV). This study was financed in part by the Coordenação de Aperfeiçoamento de Pessoal de Nível Superior – Brasil (CAPES) – Finance Code 88882.434714/ 2019-01 (EPA and LAV). This work was partially supported by the Single Cell Genomics Core Facility at the University of Nebraska-Lincoln. Confocal microscopy imaging was done in the Morrison Microscopy Core Research Facility at the University of Nebraska, Lincoln. The funders had no role in study design, data collection and analysis, decision to publish, or preparation of the manuscript.

**Competing interests:** The authors have declared that no competing interests exist.

[6–8]. FDA thus recommends including pigs for pre-clinical biology study and therapeutic testing [9,10]. Conventionally, the skin is investigated using low-content technologies such as qPCR, flow cytometry, and histology. The most recently developed single-cell RNA sequencing (scRNA-seq) technology allows simultaneously and quantitively analyzing the transcriptome of thousands of individual cells. It leads to new insights into the cell types, compositions, states, signaling, receptor-ligand interactome, and their dynamics during development, disease, and treatment [11–13]. Additionally, when combined with the high-content immunostaining and fluorescent imaging [14], or the cutting-edge Spatial Transcriptomics technology [11–13], the spatial and temporal organization of these cells and their interactions can also be obtained. scRNA-seq has been used to study rodent [15–22] and human [13,21,23–30] skins. However, using scRNA-seq to study pig skin is still rare.

In a typical scRNA-seq workflow, tissues are first dissociated into single cells (i.e., the upstream single cell preparation). These freshly isolated cells are then used to prepare RNA libraries using droplet-based technology (i.e., the downstream library preparation) [31]. The libraries are then sequenced using the deep sequencing technology [32]. Preparing high-quality single cells is critical for the success of the downstream process. Partial or incomplete tissue dissociation, cell aggregation, and cell death should be avoided during the preparation.

Recently studies showed that adding a cell preservation step between the upstream single cell preparation and downstream RNA library construction makes the scRNA-seq workflow much more flexible, manageable, and accessible to researchers [24,33–36]. First, some research facilities/institutions do not have the infrastructure and expertise to prepare scRNA-seq libraries for the freshly isolated cells. Instead, they ship cells to service facilities for RNA library construction and sequencing. Cells should be preserved during shipping. Second, most studies collect samples at multiple time points and different locations. Preparing RNA libraries at the same facilities can increase consistency and reduce experimental variations. To achieve this, cells need preservation. A suitable cell preservation method should not significantly change the cell viability, composition, and gene expression [24,33–36]. To our best knowledge, there have been no reports on how to prepare and preserve single cells from pig skin tissues for scRNA-Seq, which this study aims to address.

## Materials and methods

### Harvesting pig skin

Fresh skin tissues from healthy farm male pigs with about 30 kg of body weight were provided by the University of Nebraska-Lincoln Swine Facility. The Institutional Animal Care and Use Committee of the University of Nebraska-Lincoln granted a waiver of ethics approval for using skins harvested from dead animals. The dorsal area of the skin was washed with PBS, and the fur was removed with a disposable scalpel. The skin was disinfected with 70% ethanol, harvested using sterile scissors, and stored in the MACS Tissue Storage Solution (Miltenyi Biotech Inc) with 1% Antibiotic-Antimycotic (ThermoFisher Scientific). The samples were kept on ice and transported to the lab for single cell isolation.

### Isolating single cells

The skin was washed with sterile iced cold PBS three times, and the subcutaneous fat was scraped off using a scalpel. A full skin sample, including the epidermis and dermis, was taken with a 4-mm diameter punch. The sample was then dissociated using the Human Whole Skin Dissociation Kit (Miltenyi Biotech Inc). Briefly, 435 µL of Buffer L and 12.5 µL of Enzyme P were put into a gentleMACS C tube and gently mixed before adding 50 µL of Enzyme D and 2.5 µL of Enzyme A. One skin sample was placed into the tube and incubated at 37°C for three

hours. The tube was inverted for mixing every ten minutes. The enzymatic reaction was stopped by adding 0.5 mL ice-cold cell culture medium (DMEM + 10%FBS). The sample was then mechanically dissociated with a gentleMACS Dissociator (Miltenyi Biotech Inc) using the "h_skin_01" program. The suspension was passed through a 70 μm strainer to remove tissue debris, if any. Cells were collected by centrifugation at 300 g for 5 minutes. The supernatant was aspirated, and the cell pellet was resuspended in 5 mL DMEM + 10% FBS. Cells were counted using a Countess II FL Automated Cell Counter (ThermoFisher Scientific) on a 0.2% trypan blue staining. The viability was further assessed using the LIVE/DEAD™ Viability/Cytotoxicity Kit for mammalian cells (Invitrogen).

## Cryopreserving cells

The freshly isolated cell suspension was centrifuged at 300 g for 5 minutes. The cell pellet was resuspended with FBS containing 10% DMSO at $1x10^6$ cells/mL. Next, 1 mL cells suspension was put in a cryopreservation vial and placed in a Mr. Frosty Freezing Container (Thermo-Fisher Scientific). The container was placed in a -80°C freezer overnight before being stored in liquid $N_2$ for the long term.

## Thawing cells

The frozen vial was removed from the liquid $N_2$ storage and placed in a 37°C water bath. After thawing, the cell suspension was centrifuged at 300 g for 5 minutes and resuspended in DMEM + 10% FBS. The cell viability was assessed with the LIVE/DEAD™ Viability/Cytotoxicity Kit and flow cytometer.

## Removing dead cells and cell aggregates using Fluorescence-Activated Cell Sorting (FACS)

4 μL of 2 mM ethidium homodimer-1 was added to each milliliter cell suspension and incubated for 20 minutes at room temperature to stain the dead cells before sorting (FACSAriaII). Side-scatter and forward-scatter profiles were used to eliminate cell doublets. Living cells were gated as ethidium homodimer-1 negative. The sorted cells were re-analyzed for purity using a flow cytometer. Data were analyzed with BD FACS Diva software.

## Removing dead cells using Magnetic-Activated Cell Sorting (MACS)

An alternative approach was used to remove dead cells with the Dead Cell Remove Kit (Miltenyi Biotech Inc) following the product instruction. Briefly, the cell suspension was centrifuged at 300 g for 5 minutes. Cells were resuspended in 100 μL of Dead Cell Removal MicroBeads and incubated for 15 minutes at room temperature. Dead cells were removed using a MACS Separator (Miltenyi Biotech Inc) following the product instruction.

## Library construction

Cells were suspended in DMEM+10% FBS. The cells' density and viability were estimated using a Countess II FL Automated Cell Counter. About 8,000 cells were used as the input to generate an RNA-seq library following the 10X Genomics Chromium Next GEM Single Cell 3′ kit V3 protocol. Libraries were sequenced at Novogene using NovaSeq 6000 sequencer.

## Data processing

scRNA-seq data was processed with Cell Ranger pipeline version 4.0.0. (10x Genomics). A reference transcriptome was created by utilizing the pig reference genome (Sscrofa11.1) and its

annotation downloaded from Ensemble (https://www.ensembl.org/Sus_scrofa/Info/Index).
The reads were mapped using STAR aligner (v. 2.5.1b) against the reference transcriptome to
detect and count UMIs and expressed genes.

## Data analysis

The quality of sequencing data was assessed using quality control parameters, including the
gene counts per cell, UMI counts per cell, and mitochondrial gene expression [37,38]. These
quality control parameters were calculated as part of the scRNA-seq data analysis procedure
using the Seurat R package version 3.2.1 [37]. Since cell doublets or multiplets exhibit an aber-
rantly high gene and molecule counts, we set a maximum threshold at 4,500 genes per cell and
30,000 molecules per cell (S1 Fig). Also, we allowed cells up to 10% mitochondrial gene expres-
sion (S1 Fig).

The sample data sets were merged into one R Data object per experiment for joint cluster
analysis. Then, the UMI count data per cell were normalized and log-transformed using the
default settings of the "Normalize Data" function in Seurat [37]. Principal component analysis
(PCA) was performed using the highly variable genes for each sample. Significant principal
components were selected for subsequent cluster analysis. Single cell clustering was visualized
with uniform manifold approximation and projection (UMAP) plots with default parameters.
Cell types were annotated using marker genes with at least a 2-fold increase in individual cell
clusters compared to the remaining cells. We used CellMatch for automated cell type annota-
tion [39]. Based on the evidence-based score, we annotated the clusters by matching the identi-
fied marker genes with known cell markers in tissue-specific cell taxonomy reference
databases [39]. For single-cell trajectory analysis [40,41], we used an algorithm to learn the
changes in each cell's gene expression sequence. Once the algorithm has learned the overall
"trajectory" of gene expression changes, we placed each cell at its proper position in the trajec-
tory (line).

Statistically significant differences between cell cluster gene levels were calculated using the
MAST linear model approach, as implemented in the Seurat package. Genes were considered
as being significantly altered if gene expression levels had at least 2-fold changes and adjusted
P-values were less than 0.001 (Bonferroni correction). The overall similarity of fresh and cryo-
preserved samples' gene expression profiles was assessed by the correlation performed on
"pseudo-bulk" expression profiles [42], which were generated by summing counts together for
all cells within the same sample by using the function of "*aggregateAcross*" in the Scatter pack-
age [43]. The raw pseudo-bulk count matrices were normalized using edgeR version 3.30.3
[44]. Pearson correlation of the fresh and cryopreserved samples was computed using the nor-
malized counts. The differentially expressed pseudo-bulk genes were identified by edgeR.

## Statistical analysis

The data are presented as the mean ± S.D. We used an unpaired t-test to compare two groups
and one-way ANOVA to compare more than two groups. P-value < 0.05 was considered sta-
tistically significant. We used GraphPad Prism 6 for Windows 6.01v to perform statistical
analysis.

# Results

## Isolating and preserving single pig skin cells

We used the Miltenyi Human Whole Skin Dissociation Kit and the gentleMACS Dissociator
to dissociate the pig skin. Our results showed the combination efficiently dissociated the skin

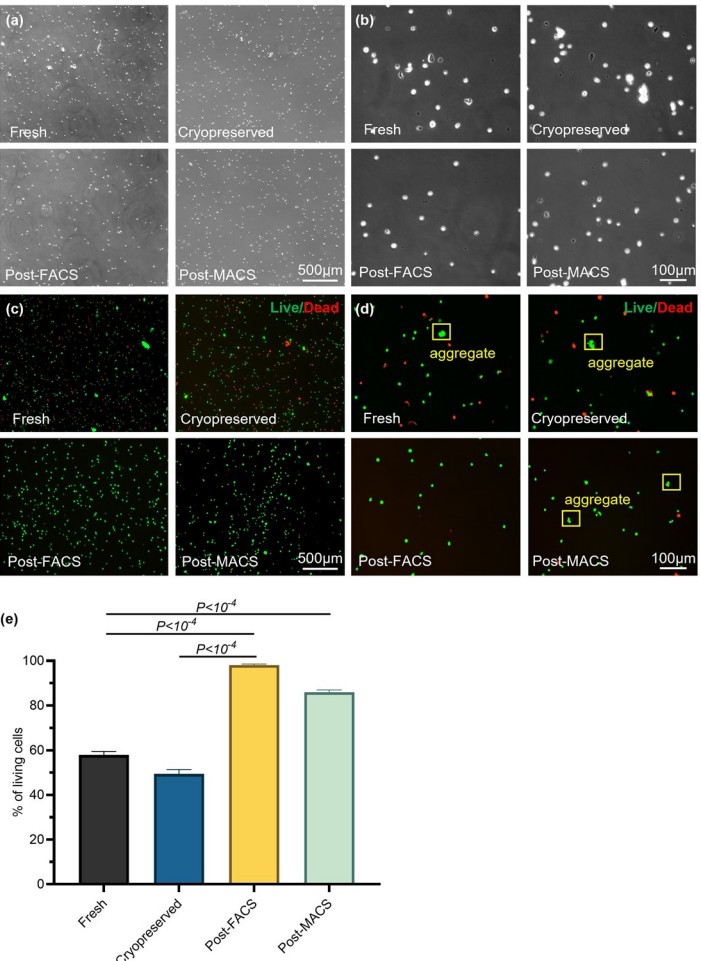

**Fig 1. Efficient single cell preparation.** Phase (a, b), live/dead staining (c, d), and flow cytometry viability quantification (e) of fresh isolated, cryopreserved, post-FACS, and post-MACS pig skin cells. n = 3 for (e).

into single cells with only a few cell aggregates and no cell or tissue debris. The resultant cells had healthy and spherical morphology (Fig 1A and 1B). Live/dead cell staining showed that most cells were live, and confirmed cell aggregates were few (Fig 1C and 1D). Flow cytometry analysis showed that >60% of cells were viable (Fig 1E). We froze the freshly isolated cells in 90% FBS + 10% DMSO at -80°C, followed by long-term storage in liquid $N_2$. After thawing, cryopreserved cells had similar spherical morphology and viability as the fresh cells (Fig 1A–1E), preliminarily indicating this method is appropriate for preserving cells. Furthermore, the cryopreservation did not induce cell aggregation.

Cell aggregates interfere with the downstream library preparation. RNAs released from dead cells negatively affect scRNA-seq results. We sought to remove both aggregated and dead cells with fluorescence-activated cell sorting (FACS) before the library preparation. Dead cells were stained with ethidium homodimer-1 dye and removed via the red fluorescence. Cell doublets were removed using the side-scatter and forward-scatter profiles. FACS quantitively removed both (Fig 1D). Since many researchers have difficulty accessing a FACS instrument, we also used an alternative and compact device, the Miltenyi magnetic-activated cell sorting (MACS), to remove the dead cells. MACS quantitively removed dead cells. However, it was not efficient to remove cell aggregates (Fig 1D). In summary, it is appropriate to combine the

Human Whole Skin Dissociation Kit, the gentleMACS Dissociator, cryopreservation, and FACS to prepare and preserve high-quality single pig skin cells for scRNA-seq.

## Cryopreservation in 90% FBS + 10% DMSO preserved gene expression

Our data showed the genes per cell, UMIs per cell, % mitochondrial genes were similar between the fresh and the cryopreserved sample (Fig 2A–2C). The pseudo-bulk expression

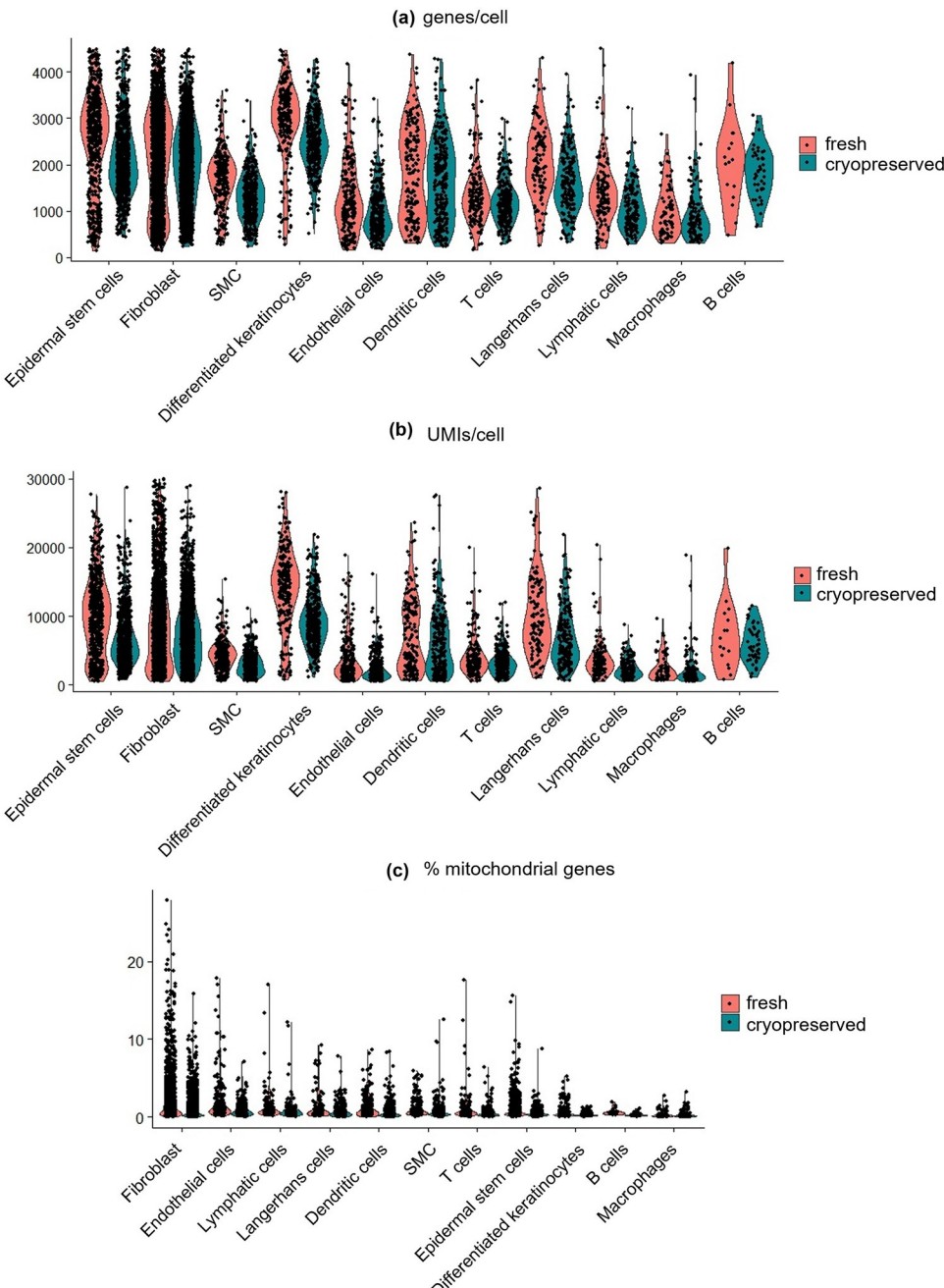

**Fig 2. Fresh and cryopreserved cells have similar quality control parameters.** The genes per cell (a), UMIs per cell (b), and % mitochondrial genes (c) in each cell of fresh and cryopreserved samples by cell type. Each dot represents one cell.

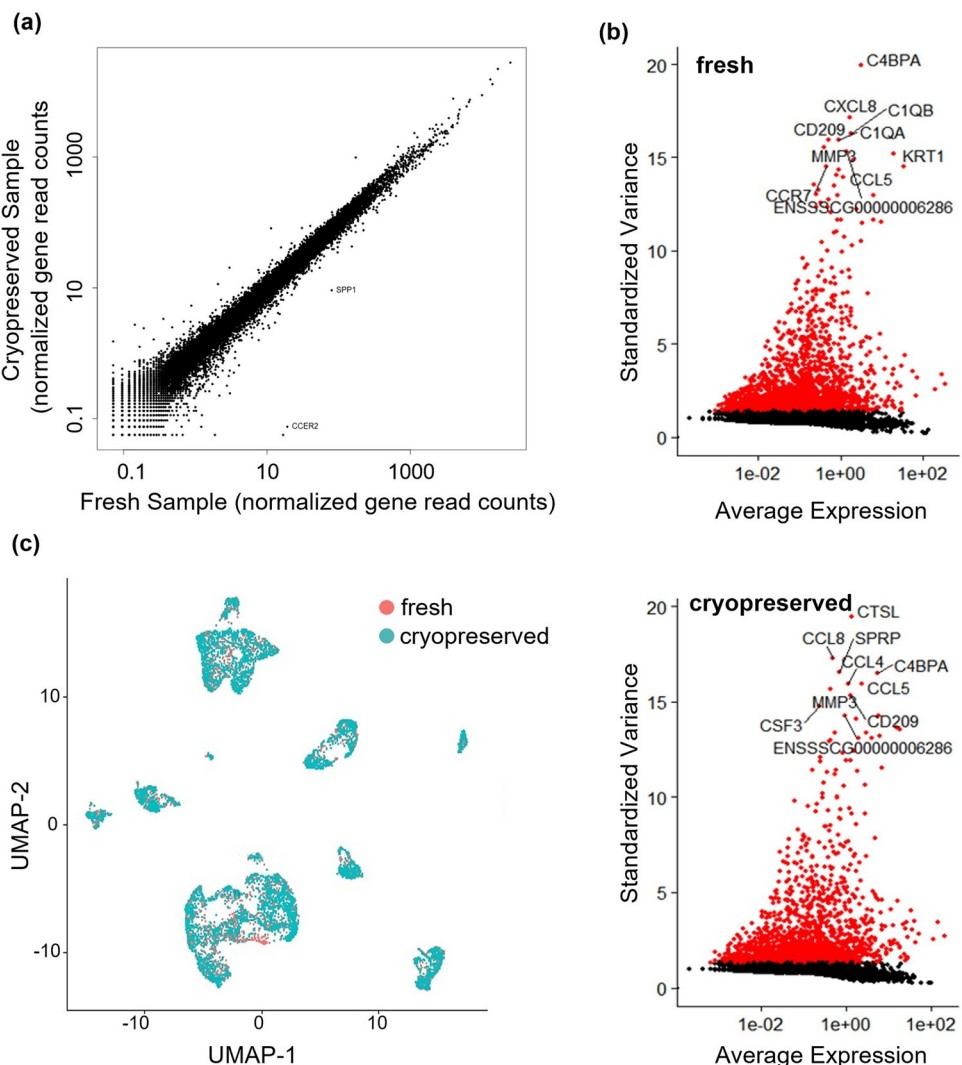

**Fig 3. Fresh and cryopreserved cells have similar gene expression profiles.** (a) The pseudo-bulk expression profiles of fresh and cryopreserved cells are compared using correlation scatter plots. The profiles correlate well (R = 0.981). (b) Analysis of the highly variable genes shows fresh and cryopreserved samples share the highly variable features with similar variance and average expression. (c) UAMP shows fresh and cryopreserved samples have similar structures.

profiles of cryopreserved and fresh cells were compared using correlation analysis to assess if cryopreservation alters the gene expression. The expression profile of the cryopreserved sample correlated very well with this of fresh cells (R = 0.981) (Fig 3A). Among 20,428 expressed genes, there were only 57 differentially expressed genes (DEGs) between fresh and cryopreserved samples with FDR adjusted P-values < 0.01. This indicates the gene expression profiles between fresh and cryopreserved samples are almost the same. Next, we analyzed the subset of genes that exhibited high cell-to-cell variation in the samples (Fig 3B). Fresh and cryopreserved samples shared the highly variable features (e.g., CD209, MMP3, CCL5, C4BPA) with similar variance and average expression. Lastly, we assembled the fresh and the cryopreserved sample into an integrated reference and visualized the integration using a non-linear dimensional reduction UMAP (Fig 3C). Again, the fresh and cryopreserved cells formed similar structures. These results show that cryopreservation in 90% FBS + 10% DMSO did not alter the gene expression significantly.

## Cryopreservation in 90% FBS + 10% DMSO retained the major skin cell types

Next, we sought to answer if the single cell preparation protocol could retain the major skin cell types and if the cryopreservation changed the cell composition. We performed a clustering analysis of the integrated cryopreserved and fresh cells using a graph-based clustering approach [37,45]–the Louvain algorithm [46]. This method embeds cells in a graph structure, clusters cells with similar feature expressions, and partitions the graph into highly interconnected communities. We identified 18 clusters (S2 Fig) and the top 10 marker genes for each cluster (S3 Fig). We then used a mixed strategy to annotate the clusters. We used CellMatch [46] to annotate clusters based on the evidence-based score by matching the identified marker genes with known cell markers in tissue-specific cell taxonomy reference databases. Also, we used cell marker genes from recent scRNA-seq studies of skin [16,27] to identify the remaining cell types. This allows us to identify the major skin cell types (Fig 4A). The expression levels for the markers of each cell type were shown in Fig 5 and S1 Table. The top 10 marker genes for each cell type were also identified (S4 Fig). We compared the fresh and the cryopreserved samples regarding the relative organization and connection between cell clusters (Fig 4A). The two

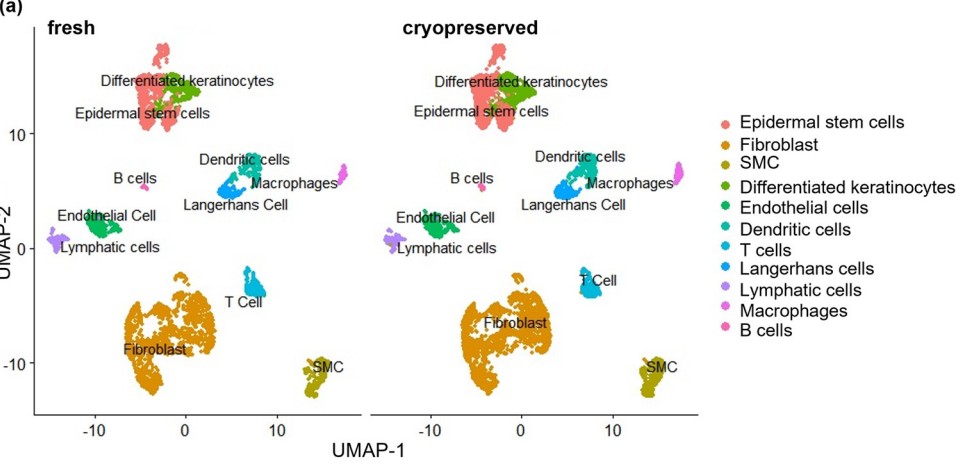

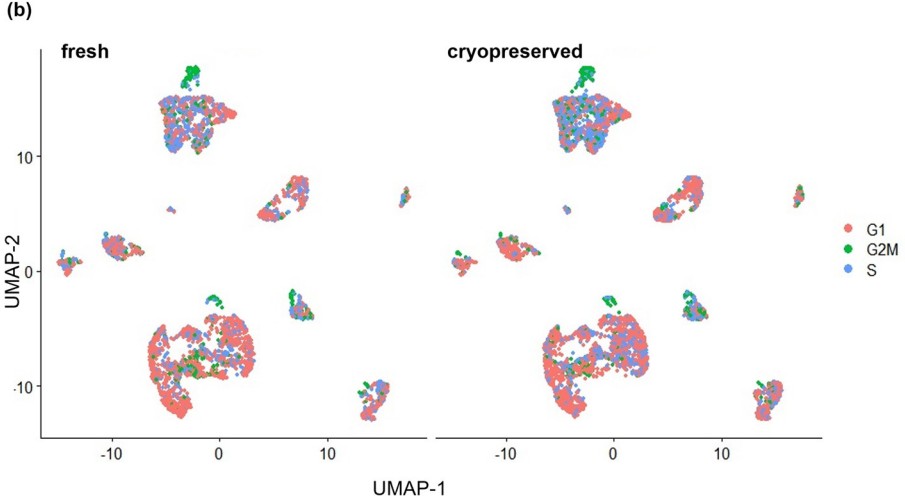

**Fig 4.** Fresh and cryopreserved samples have similar cell types (a) and cell cycle status (b).

**Table 1. Cellular composition in fresh and cryopreserved samples.**

| Cell type | cryopreserved | fresh |
|---|---|---|
| Epidermal stem cells | 17.8% | 16.0% |
| Fibroblast | 37.9% | 45.5% |
| SMC | 8.2% | 7.2% |
| Differentiated keratinocytes | 7.7% | 6.7% |
| Endothelial cells | 6.3% | 6.5% |
| Dendritic cells | 5.9% | 4.6% |
| T cells | 5.5% | 4.3% |
| Langerhans cells | 4.1% | 3.1% |
| Lymphatic cells | 2.9% | 3.3% |
| Macrophages | 2.5% | 2.1% |
| B cells | 0.9% | 0.4% |

samples were very similar. Also, the cryopreservation did not significantly alter the cell compositions (Table 1).

## Cryopreservation retained the cell cycle status and differentiation trajectory

Further, we used cell cycle regression to analyze the effect of cryopreservation on the cell cycle status. We identified the cell cycle heterogeneity from cryopreserved and fresh samples by calculating cell cycle phase scores based on canonical markers [39]. Our results showed no differences between fresh and cryopreserved keratinocytes regarding G2/M-phase genes (Fig 4B). Using the single-cell trajectory analysis [40,41], we found that epidermal stem cells differentiated into keratinocytes in both fresh and cryopreserved samples (Fig 6A and 6B). The results indicate the data quality for both the fresh and cryopreserved cells is sufficient for transcriptional trajectory analysis.

## Discussion

Preparing high-quality single cells is a critical step for a successful scRNA-seq study. A suitable dissociation method should dissolve the whole skin tissue, including epidermis and dermis, to

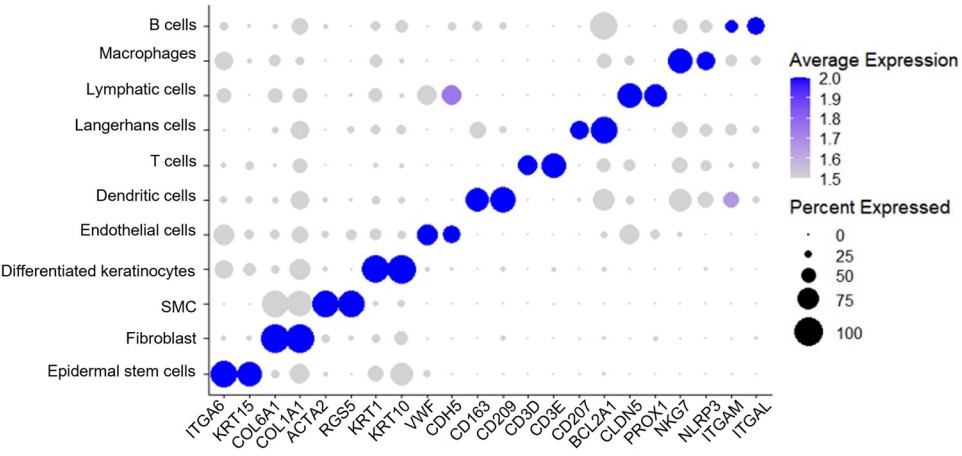

**Fig 5. Dot plot of marker genes for pig skin cells.**

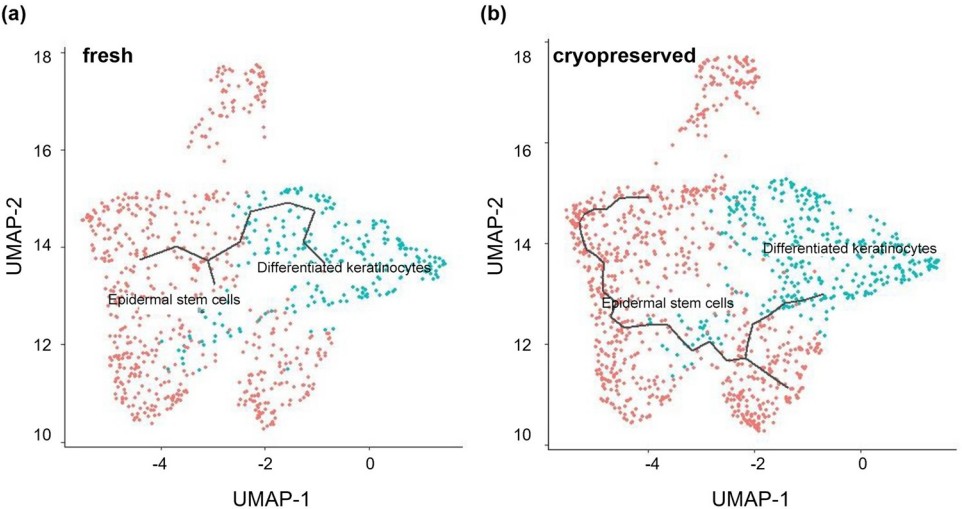

**Fig 6. Single-cell trajectory analysis shows epidermal stem cells differentiate into keratinocytes in both fresh and cryopreserved samples.**

release most if not all cells [23,47,48]. The resultant cells should have high viability and few aggregates. A dissociation method with high cell yield and cell viability can minimize skin samples and allow detecting cell types with small numbers. A few protocols have been published for preparing single cells from rodent [49–55] and human skins [56–61] via enzymatic digestion and mechanical dissociation. These methods vary in enzyme types and concentrations, digestion time, and temperature [49–61]. Most of them use manual mechanical dissociation, such as dissection, mincing, agitation, pipetting, or passing through syringes that are inefficient and inconsistent. Consequently, the dissociation efficiency and the resultant cell viability, composition vary between methods, publications, and batches. The variations create challenges when comparing scRNA-seq results from different labs. Thus, there is a critical need to develop a standardized single cell preparation method. A standard procedure could be best achieved using validated, commercially available enzyme kits and automated mechanical dissociation devices.

Although we could not find a commercial kit designed explicitly for pig skin, the Miltenyi Human Whole Skin Dissociation Kit and the gentleMACS Dissociator had been used to prepare human skin single cells for scRNA-seq [23,24,27,28,62,63]. Due to the similarity of the human and pig skins, we hypothesized that they could also be used to prepare single pig skin cells. With our protocol, the skin tissue was close to completely dissolved. Only small numbers of cell aggregates were found, and they could be robustly removed with FACS (Fig 1). We were able to isolate $6.0–7.5\times10^4$ viable single cells with a 4-mm full skin biopsy punch. These cells are sufficient for a complete scRNA-seq flow, including cell counting, cell viability, quality assessment, RNA library construction, and quality control.

Reliable methods for preserving single cells make the scRNA-Seq workflow much more flexible and manageable [24,33–36]. A few methods, including cryopreservation in medium containing DMSO, using commercial cell preservation reagent, or methanol fixing, have been reported to preserve single cells [24,33,35,36]. DMSO minimizes forming large intracellular ice crystals, which damage and kill cells. Methanol fixing works through dehydrating cells to preserve nucleic acids in a collapsed form at high concentrations. Upon rehydration, nucleic acids can be restored to their original form and harvested for library preparation [62,64].

Literature research showed the methanol fixing method result in high ambient RNA background and a lower gene expression correlation to un-preserved cells [33]. The same was true for when using commercial CellCover reagent [33]. On the other hand, cryopreserving human and rodent primary cells or cell lines using DMSO plus FBS did not reduce the cell viability and alter the cell composition and gene expression significantly [24,33–36,65]. Our results with pig skin cells (Figs 1–6) agreed well with these literature studies.

In summary, we showed that high-quality single pig skin cells could be generated using the Miltenyi Biotec Whole Skin Dissociation kit and Gentle MACS Dissociator. Single cells could be cryopreserved in 90% FBS+10% DMSO, and the cryopreservation did not significantly alter gene expression and cell compositions. Dead cells and cell aggregates could be removed via FACS before preparing libraries. Using these methods, we obtained high-quality scRNA-Seq data to identify the major skin cell types. The limitation of this study is that only healthy pig skins from young pigs were used. Future studies should test if the protocol works for aged or diseased pig skins.

## Supporting information

**S1 Fig. Quality control parameters of fresh and cryopreserved (frozen) pig skin cells, including the number of genes, UMIs, and % of mitochondrial gene in each cell.** (a) pre-cut-off and (b) post-cut-off data are shown. 4500 genes/cell, 30,000 molecules/cell, and 10% mitochondrial genes are set as the maximum threshold to exclude the doublets, multiples, and low-quality single cells. Each dot represents one cell.
(TIF)

**S2 Fig.** Clustering analysis of combined (a) and separated (b) fresh and cryopreserved samples shows that the two samples have similar clusters.
(TIF)

**S3 Fig. Top 10 marker genes for each cluster.**
(TIF)

**S4 Fig. Top 10 marker genes for each cell type.**
(TIF)

**S1 Table. Markers used for annotating cell clusters.**
(TIF)

## Author Contributions

**Conceptualization:** Mark A. Carlson, William H. Velander, Yuguo Lei.

**Data curation:** Li Han, Carlos P. Jara, Rafał K. Wóycicki.

**Funding acquisition:** Mark A. Carlson, William H. Velander, Marc Libault, Yuguo Lei.

**Investigation:** Li Han, Carlos P. Jara, Ou Wang, Yu Shi, Xinran Wu, Sandra Thibivilliers.

**Project administration:** Yuguo Lei.

**Supervision:** William H. Velander, Eliana P. Araújo, Marc Libault, Chi Zhang, Yuguo Lei.

**Writing – original draft:** Li Han.

**Writing – review & editing:** Marc Libault, Chi Zhang, Yuguo Lei.

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
