## [Decision Letter · Decision Letter 0]

13 May 2021

PONE-D-21-09015

Isolating and Cryo-Preserving Pig Skin Cells for Single Cell RNA Sequencing Study

PLOS ONE

Dear Dr. Lei,

Thank you for submitting your manuscript to PLOS ONE. After careful consideration, we feel that it has merit but does not fully meet PLOS ONE’s publication criteria as it currently stands. Therefore, we invite you to submit a revised version of the manuscript that addresses the points raised during the review process.

We look forward to receiving your revised manuscript.

Kind regards,

Nazmul Haque

Academic Editor

PLOS ONE

Journal Requirements:

2. We note that you are reporting an analysis of a microarray, next-generation sequencing, or deep sequencing data set. PLOS requires that authors comply with field-specific standards for preparation, recording, and deposition of data in repositories appropriate to their field. Please upload these data to a stable, public repository (such as ArrayExpress, Gene Expression Omnibus (GEO), DNA Data Bank of Japan (DDBJ), NCBI GenBank, NCBI Sequence Read Archive, or EMBL Nucleotide Sequence Database (ENA)). In your revised cover letter, please provide the relevant accession numbers that may be used to access these data. For a full list of recommended repositories, see http://journals.plos.org/plosone/s/data-availability#loc-omics or http://journals.plos.org/plosone/s/data-availability#loc-sequencing.

Reviewers' comments:

Reviewer's Responses to Questions

**Comments to the Author**

1. Is the manuscript technically sound, and do the data support the conclusions?

Reviewer #1: Yes

Reviewer #2: Partly

2. Has the statistical analysis been performed appropriately and rigorously? 

Reviewer #1: Yes

Reviewer #2: Yes

3. Have the authors made all data underlying the findings in their manuscript fully available?

Reviewer #1: Yes

Reviewer #2: Yes

4. Is the manuscript presented in an intelligible fashion and written in standard English?

Reviewer #1: Yes

Reviewer #2: No

5. Review Comments to the Author

Reviewer #1: In this study, Han et al describe a method for isolating and preserving primary porcine skin cells for subsequent analysis by single cell RNA sequencing. They demonstrate that the cryopreservation process o purified cells minimally impacts cell viability and the transcriptional profile of the isolated cells.

Comments:

1) There are some typos in Figure 1B that should be corrected (Cry-thaw, FACs, MACs)

2) Figure 4 and Figure 5 are very duplicative. The figures could be combined, or figure 4 moved to supplemental data (or deleted).

3) Figure 3a: the axis of the plot need to be labeled more clearly. Assuming that the data shown is the relative gene expression. Log transforming the data on such a plot would be a more conventional way to display the data, and highlighting some important genes would help with orientation.

4) The description of the trajectory analysis could use some refinement. As currently written the data is presented as demonstrating that cell differentiation occurs in both fresh and frozen cells. A more correct/helpful description of the data would be something along the lines of "the data quality obtained from both fresh and frozen cells was of sufficient quality to allow transcriptional trajectory/pseudotime analysis"

5) Population identification/annotation is a very important step in scRNAseq analysis, especially for novel tissue types and "non-standard" model systems. It might be helpful to move some of the linage-defining gene expression data from supplemental data to a primary figure, and perhaps use a dot plot to highlight key genes from each population.

Reviewer #2: The current manuscript entitled ‘Isolating and Cryo-Preserving Pig Skin Cells for Single Cell RNA Sequencing Study’ is well written. However, there are some points need to be addressed before publication –

1# If I want to summarize the content of this manuscript, there would be three key points – i) Isolation of single cell from Pig skin, ii) Cryo-preservation of that single cell and iii) RNA sequencing of that single pig skin cell. scRNA-seq is a common and established method of gene expression analysis if the single cell is available. Thus, the main part of this manuscript is the isolation and preservation of single pig skin cells. Unfortunately, the authors highlighted scRNA-seq in the abstract not the isolation methodology. In the Introduction section the authors did the same approach. They need to rewrite the Abstract and Introduction in a more presentable format. Additionally, the last paragraph of Introduction needs to be removed. Introduction should not contain any methodology or results. Instate of that the authors need to present their aim of this research.

2# Which one is correct – pigskin or pig skin? Authors need to follow only one style throughout the manuscript.

3# In the Materials and Methods, the authors claimed only Methods. Does it compatible with Plos One? The first sub-heading in this section is entitled as ‘Pig skins’. Does it mean anything? It could be ‘Harvesting of pig skin’. Writing of a manuscript and writing of lab note is different. Authors are suggested to recheck all the sub-headings in this manuscript and change them in the more presentable format, if required. Additionally, they are suggested to follow the international format of writing the company names for any chemicals and instruments throughout the manuscript.

4# The isolation of single cell from pig skin is the main part of this manuscript. So, authors need to describe in detail procedure in the Materials and Methods. The present description is not satisfactory.

5# Author wrote ‘Libraries were sequenced using Illumina technology.’ Does they have sequencer in their Lab? If so, then provide the company name. If not, then mention the commercial service name.

6# Authors need to recheck the suitability of data deposition statement in the body text. If nor, they are requested to remove the ‘Data and code availability’ section from the text.

7# Which one is correct – liquid N2 or liquid N2? Authors need to follow only one style throughout the manuscript. Furthermore, they need to check for cryopreservation or cryo-preservation. And so on ……….

8# It is well established that 10% DMSO provides good preservation of live cells. The authors of this manuscript also used the same procedure. However, they want to claim that they did something new. They are requested to show every point they have modified in the general procedure. Additionally, several times they used the term ‘DMSO-based cryopreservation’. For example, they claim that DMSO-based cryopreservation preserved gene expression. What does it mean? Does it mean that among several cryopreservation techniques DMSO-based cryopreservation showed optimum result? Does it mean that DMSO-based cryopreservation is mandatory for pig skin cell? The authors need to show a comparative analysis among different preservation techniques if they want to claim so. Additionally, they need to provide a positive control (either human skin/rodent skin cells) to compare.

9# In summary, the authors isolated single cells from pig skin, cryopreserved and compare it with fresh cell in genomic level. Thus, the merit goes to the impact of preservation in genomic level. So far, this manuscript is pioneer to isolating single cells from pig skin. In this regards it has novelty. But if they want to show the impact of preservation then the present format of manuscript is not suitable. They need to add positive control and comparison data for different preservation techniques. And rewrite the manuscript in particular manner.

10# Rigorous English editing is required.

6. PLOS authors have the option to publish the peer review history of their article (what does this mean?). If published, this will include your full peer review and any attached files.

Reviewer #1: **Yes: **Adam T Waickman

Reviewer #2: No

---

## [Author Response · Author response to Decision Letter 0]

16 Sep 2021

Reviewer #1: 

1) There are some typos in Figure 1B that should be corrected (Cry-thaw, FACs, MACs)

 Response: Revised. 

2) Figure 4 and Figure 5 are very duplicative. The figures could be combined, or figure 4 moved to supplemental data (or deleted).

 Response: We moved Fig 4 to supplemental data (S2 Fig).

3) Figure 3a: the axis of the plot need to be labeled more clearly. Assuming that the data shown is the relative gene expression. Log transforming the data on such a plot would be a more conventional way to display the data, and highlighting some important genes would help with orientation.

 Response: Revised following the suggestion. 

4) The description of the trajectory analysis could use some refinement. As currently written the data is presented as demonstrating that cell differentiation occurs in both fresh and frozen cells. A more correct/helpful description of the data would be something along the lines of "the data quality obtained from both fresh and frozen cells was of sufficient quality to allow transcriptional trajectory/pseudotime analysis"

 Response: Revised as suggested. 

5) Population identification/annotation is a very important step in scRNAseq analysis, especially for novel tissue types and "non-standard" model systems. It might be helpful to move some of the linage-defining gene expression data from supplemental data to a primary figure, and perhaps use a dot plot to highlight key genes from each population.

 Response: Following the suggestion, we added Fig 5. 

Reviewer #2: 

1# If I want to summarize the content of this manuscript, there would be three key points – i) Isolation of single cell from Pig skin, ii) Cryo-preservation of that single cell and iii) RNA sequencing of that single pig skin cell. scRNA-seq is a common and established method of gene expression analysis if the single cell is available. Thus, the main part of this manuscript is the isolation and preservation of single pig skin cells. Unfortunately, the authors highlighted scRNA-seq in the abstract not the isolation methodology. In the Introduction section the authors did the same approach. They need to rewrite the Abstract and Introduction in a more presentable format. Additionally, the last paragraph of introduction needs to be removed. Introduction should not contain any methodology or results. Instate of that the authors need to present their aim of this research.

 Response: Thanks for the comments. We revised the manuscript as suggested. 

2# Which one is correct – pigskin or pig skin? Authors need to follow only one style throughout the manuscript.

 Response: Pig skin is used in the revised manuscript. 

3# In the Materials and Methods, the authors claimed only Methods. Does it compatible with Plos One? The first sub-heading in this section is entitled as ‘Pig skins’. Does it mean anything? It could be ‘Harvesting of pig skin’. Writing of a manuscript and writing of lab note is different. Authors are suggested to recheck all the sub-headings in this manuscript and change them in the more presentable format, if required. Additionally, they are suggested to follow the international format of writing the company names for any chemicals and instruments throughout the manuscript.

 Response: Revised as suggested.

4# The isolation of single cell from pig skin is the main part of this manuscript. So, authors need to describe in detail procedure in the Materials and Methods. The present description is not satisfactory.

 Response: We added details of the isolation process. 

5# Author wrote ‘Libraries were sequenced using Illumina technology.’ Does they have sequencer in their Lab? If so, then provide the company name. If not, then mention the commercial service name.

 Response: Revised as suggested. 

6# Authors need to recheck the suitability of data deposition statement in the body text. If nor, they are requested to remove the ‘Data and code availability’ section from the text.

 Response: Removed from the text.

7# Which one is correct – liquid N2 or liquid N2? Authors need to follow only one style throughout the manuscript. Furthermore, they need to check for cryopreservation or cryo-preservation. And so on ……….

 Response: Changed all the “liquid N2” to “liquid N2”. In addition, we replaced “frozen” by “cryopreserved”.

8# It is well established that 10% DMSO provides good preservation of live cells. The authors of this manuscript also used the same procedure. However, they want to claim that they did something new. They are requested to show every point they have modified in the general procedure. Additionally, several times they used the term ‘DMSO-based cryopreservation’. For example, they claim that DMSO-based cryopreservation preserved gene expression. What does it mean? Does it mean that among several cryopreservation techniques DMSO-based cryopreservation showed optimum result? Does it mean that DMSO-based cryopreservation is mandatory for pig skin cell? The authors need to show a comparative analysis among different preservation techniques if they want to claim so. Additionally, they need to provide a positive control (either human skin/rodent skin cells) to compare.

 Response: Thanks for the comments.

 First, we changed the “DMSO-based cryopreservation” to “cryopreservation in 90% FBS +10% DMSO” to avoid the confusion. 

Second, we did not claim: 1) we invented a new cryopreservation method, or improved/modified the DMSO-based cryopreservation method; 2) the DMSO-based cryopreservation is better than other cell preservation methods. 

The goal of this work is to find, demonstrate, validate a robust method/protocol for isolating and preserving single pig skin cells for scRNA-Seq study. To our best knowledge, there have been no reports on how to prepare and preserve single cells from pig skin tissue for scRNA-Seq in the literature. We showed that using our reported method/protocol, high-quality scRNA-Seq data could be generated to identify the major skin cell types. We believe this report is very valuable for the skin research scientific community.

9# In summary, the authors isolated single cells from pig skin, cryopreserved and compare it with fresh cell in genomic level. Thus, the merit goes to the impact of preservation in genomic level. So far, this manuscript is pioneer to isolating single cells from pig skin. In this regards it has novelty. But if they want to show the impact of preservation then the present format of manuscript is not suitable. They need to add positive control and comparison data for different preservation techniques. And rewrite the manuscript in particular manner.

 Response: same as 8#.

10# Rigorous English editing is required.

 Response: Revised as the suggestion.

---

## [Decision Letter · Decision Letter 1]

13 Dec 2021

PONE-D-21-09015R1Isolating and cryopreserving pig skin cells for single-cell RNA sequencing studyPLOS ONE

Dear Dr. Lei,

Thank you for submitting your manuscript to PLOS ONE. After careful consideration, we feel that it has merit but does not fully meet PLOS ONE’s publication criteria as it currently stands. Therefore, we invite you to submit a revised version of the manuscript that addresses the points raised during the review process. Specifically, If the methods have not been developed by you then please refer to the article(s) from where you have adopted or modified these methods.  Please submit your revised manuscript by Jan 27 2022 11:59PM. If you will need more time than this to complete your revisions, please reply to this message or contact the journal office at plosone@plos.org. Please include the following items when submitting your revised manuscript:A rebuttal letter that responds to each point raised by the academic editor and reviewer(s). You should upload this letter as a separate file labeled 'Response to Reviewers'.A marked-up copy of your manuscript that highlights changes made to the original version. You should upload this as a separate file labeled 'Revised Manuscript with Track Changes'.An unmarked version of your revised paper without tracked changes. You should upload this as a separate file labeled 'Manuscript'.If applicable, we recommend that you deposit your laboratory protocols in protocols.io to enhance the reproducibility of your results. Protocols.io assigns your protocol its own identifier (DOI) so that it can be cited independently in the future. For instructions see: https://journals.plos.org/plosone/s/submission-guidelines#loc-laboratory-protocols. Additionally, PLOS ONE offers an option for publishing peer-reviewed Lab Protocol articles, which describe protocols hosted on protocols.io. Read more information on sharing protocols at https://plos.org/protocols?utm_medium=editorial-email&utm_source=authorletters&utm_campaign=protocols.

We look forward to receiving your revised manuscript.

Kind regards,

Nazmul Haque

Academic Editor

PLOS ONE

Journal Requirements:

Reviewers' comments:

Reviewer's Responses to Questions

**Comments to the Author**

1. If the authors have adequately addressed your comments raised in a previous round of review and you feel that this manuscript is now acceptable for publication, you may indicate that here to bypass the “Comments to the Author” section, enter your conflict of interest statement in the “Confidential to Editor” section, and submit your "Accept" recommendation.

Reviewer #1: All comments have been addressed

Reviewer #2: (No Response)

2. Is the manuscript technically sound, and do the data support the conclusions?

Reviewer #1: Yes

Reviewer #2: (No Response)

3. Has the statistical analysis been performed appropriately and rigorously? 

Reviewer #1: Yes

Reviewer #2: (No Response)

4. Have the authors made all data underlying the findings in their manuscript fully available?

Reviewer #1: Yes

Reviewer #2: (No Response)

5. Is the manuscript presented in an intelligible fashion and written in standard English?

Reviewer #1: Yes

Reviewer #2: (No Response)

6. Review Comments to the Author

Reviewer #1: (No Response)

Reviewer #2: A) In response to my query #8 the authors answered as follows –

1) We invented a new cryopreservation method, or improved/modified the DMSO-based cryopreservation method;

Observation: The authors used three keywords invention, improvement and modification. Which one is appropriate? Do they really invent a new protocol??? I think they just modified the existing protocol for pig skin cell isolation. If so, then they need to let us know in which points they have changed. Furthermore, does the modification bring better result? If so, then they need to show us both results in a comparison manner. We need to understand that their modification is meaningful.

2) The DMSO-based cryopreservation is better than other cell preservation methods.

Observation: We all agree with this statement. Everybody knows that the DMSO-based cryopreservation method is the best one. Then, what is new here the author is claiming. Why the authors are highlighting DMSO-based cryopreservation for pig cells? This would be a normal procedure for cell preservation.

B) The reviewer is not satisfied with the answer for #9. The authors need to organize their manuscript in a manner that they just isolated single cells from pig skin.

7. PLOS authors have the option to publish the peer review history of their article (what does this mean?). If published, this will include your full peer review and any attached files.

Reviewer #1: **Yes: **Adam Waickman

Reviewer #2: No

---

## [Author Response · Author response to Decision Letter 1]

19 Jan 2022

Reviewers' comments:

Reviewer #2: A) In response to my query #8 the authors answered as follows –

1) We invented a new cryopreservation method, or improved/modified the DMSO-based cryopreservation method;

Observation: The authors used three keywords invention, improvement and modification. Which one is appropriate? Do they really invent a new protocol??? I think they just modified the existing protocol for pig skin cell isolation. If so, then they need to let us know in which points they have changed. Furthermore, does the modification bring better result? If so, then they need to show us both results in a comparison manner. We need to understand that their modification is meaningful.

2) The DMSO-based cryopreservation is better than other cell preservation methods.

Observation: We all agree with this statement. Everybody knows that the DMSO-based cryopreservation method is the best one. Then, what is new here the author is claiming. Why the authors are highlighting DMSO-based cryopreservation for pig cells? This would be a normal procedure for cell preservation.

Response: There is some miscommunication here. We strongly agree with the reviewer about the observations. In our response to to query #8, we said: 

“Second, we did not claim: 1) we invented a new cryopreservation method, or improved/modified the DMSO-based cryopreservation method; 2) the DMSO-based cryopreservation is better than other cell preservation methods”

We said “we did not claim”. 

B) The reviewer is not satisfied with the answer for #9. The authors need to organize their manuscript in a manner that they just isolated single cells from pig skin.

Response: Sorry that we did not have a clear answer in our last response. We did re-ogranize the manuscript in a manner suggested by the reviewer in the previous revision.

Again, thanks for all the suggestions!

---

## [Decision Letter · Decision Letter 2]

31 Jan 2022

Isolating and cryopreserving pig skin cells for single-cell RNA sequencing study

PONE-D-21-09015R2

Dear Dr. Lei,

We’re pleased to inform you that your manuscript has been judged scientifically suitable for publication and will be formally accepted for publication once it meets all outstanding technical requirements.

Kind regards,

Nazmul Haque

Academic Editor

PLOS ONE

Additional Editor Comments (optional):

Reviewers' comments:

Reviewer's Responses to Questions

**Comments to the Author**

1. If the authors have adequately addressed your comments raised in a previous round of review and you feel that this manuscript is now acceptable for publication, you may indicate that here to bypass the “Comments to the Author” section, enter your conflict of interest statement in the “Confidential to Editor” section, and submit your "Accept" recommendation.

Reviewer #2: (No Response)

2. Is the manuscript technically sound, and do the data support the conclusions?

Reviewer #2: (No Response)

3. Has the statistical analysis been performed appropriately and rigorously? 

Reviewer #2: (No Response)

4. Have the authors made all data underlying the findings in their manuscript fully available?

Reviewer #2: (No Response)

5. Is the manuscript presented in an intelligible fashion and written in standard English?

Reviewer #2: (No Response)

6. Review Comments to the Author

Reviewer #2: (No Response)

7. PLOS authors have the option to publish the peer review history of their article (what does this mean?). If published, this will include your full peer review and any attached files.

Reviewer #2: No

---

## [Editor Report · Acceptance letter]

8 Feb 2022

PONE-D-21-09015R2 

Isolating and cryopreserving pig skin cells for single-cell RNA sequencing study 

Dear Dr. Lei:

I'm pleased to inform you that your manuscript has been deemed suitable for publication in PLOS ONE. Congratulations! Your manuscript is now with our production department. 

Kind regards, 

on behalf of

Dr. Nazmul Haque 

Academic Editor

PLOS ONE